# The Effects of Dual GLP-1/GIP Receptor Agonism on Glucagon Secretion—A Review

**DOI:** 10.3390/ijms20174092

**Published:** 2019-08-22

**Authors:** David S. Mathiesen, Jonatan I. Bagger, Natasha C. Bergmann, Asger Lund, Mikkel B. Christensen, Tina Vilsbøll, Filip K. Knop

**Affiliations:** 1Center for Clinical Metabolic Research, Gentofte Hospital, University of Copenhagen, 2900 Hellerup, Denmark; 2Steno Diabetes Center Copenhagen, 2820 Gentofte, Denmark; 3Department of Clinical Pharmacology, Bispebjerg Hospital, University of Copenhagen, 2400 Copenhagen, Denmark; 4Department of Clinical Medicine, Faculty of Health and Medical Sciences, University of Copenhagen, 2200 Copenhagen, Denmark; 5Novo Nordisk Foundation Center for Basic Metabolic Research, Faculty of Health and Medical Sciences, University of Copenhagen, 2200 Copenhagen, Denmark

**Keywords:** glucagon, glucagon-like peptide 1, glucose-dependent insulinotropic polypeptide, gastric inhibitory peptide, incretins, dual-agonism, type 2 diabetes, obesity

## Abstract

The gut-derived incretin hormones glucagon-like peptide 1 (GLP-1) and glucose-dependent insulinotropic polypeptide (GIP) are secreted after meal ingestion and work in concert to promote postprandial insulin secretion. Furthermore, GLP-1 inhibits glucagon secretion when plasma glucose concentrations are above normal fasting concentrations while GIP acts glucagonotropically at low glucose levels. A dual incretin receptor agonist designed to co-activate GLP-1 and GIP receptors was recently shown to elicit robust improvements of glycemic control (mean haemoglobin A1c reduction of 1.94%) and massive body weight loss (mean weight loss of 11.3 kg) after 26 weeks of treatment with the highest dose (15 mg once weekly) in a clinical trial including overweight/obese patients with type 2 diabetes. Here, we describe the mechanisms by which the two incretins modulate alpha cell secretion of glucagon, review the effects of co-administration of GLP-1 and GIP on glucagon secretion, and discuss the potential role of glucagon in the therapeutic effects observed with novel unimolecular dual GLP-1/GIP receptor agonists. For clinicians and researchers, this manuscript offers an understanding of incretin physiology and pharmacology, and provides mechanistic insight into future antidiabetic and obesity treatments.

## 1. Introduction

The discovery of insulin in the dawn of the 20th century [1] also led to the discovery of a hyperglycemic substance originating from the pancreas [2]. This substance was named glucagon, short for glucose agonist, 30 years before it was crystallized and its amino acid sequence was determined [3,4]. Initially, glucagon was exclusively perceived as a hormone opposing the effect of insulin on blood glucose by stimulating endogenous glucose production, but years of research have paved the way for a broader understanding of glucagon biology. Today, it is generally accepted that glucagon also affects regulation of amino acid metabolism [5], while supraphysiological levels of glucagon affect energy intake and body weight [6,7], energy expenditure [8,9], as well as lipid metabolism [10,11] in humans. Furthermore, exogenous glucagon has positive chronotropic and inotropic effects on the human heart [12,13,14]. Importantly, hypersecretion of glucagon and ensuing stimulation of endogenous glucose production plays an important role in the pathophysiology of diabetes contributing to the hyperglycemia characterizing the diabetic state [15,16,17]. Glucose, insulin and somatostatin exert inhibitory effects on alpha cell secretion of glucagon, whereas hypoglycemia and amino acids are known to stimulate the secretion of glucagon. Also, the gut-derived incretin hormones glucagon-like peptide 1 (GLP-1) and glucose-dependent insulinotropic polypeptide (GIP), best known for their potent insulinotropic effects, modulate glucagon secretion and they may play a role in the dysregulation of glucagon secretion observed in patients with type 2 diabetes [16]. GLP-1 inhibits glucagon secretion during hyperglycemia, but not when glucose levels return to euglycemia or during hypoglycemia [18]. As a result, GLP-1 receptor (GLP-1R) agonists (GLP-1RA) reduce hyperglycemia with little risk of hypoglycemic episodes. In non-diabetics, GIP stimulates glucagon secretion during hypoglycemia and potentiates insulin secretion during hyperglycemia [19]. The insulinotropic effect of GIP is severely reduced in patients with type 2 diabetes [20], while the glucagonotropic effect of GIP seems to prevail [21]. Therefore, GIP receptor (GIPR) agonism has not been pursued in the treatment of type 2 diabetes, but novel GLP-1/GIP receptor agonists may soon redefine the pharmacological role of GIP. Relevant literature for this review was retrieved from searches in the electronic PubMed database using the following search terms: “incretin”, “GLP-1”, “GIP”, “glucagon” and “dual agonist”. Furthermore, manual reference searches in relevant papers and abstracts from scientific meetings were performed.

This review provides insights into the mechanisms by which GLP-1 and GIP modulate glucagon secretion and discusses findings on glucagon in the context of GLP-1 and GIP co-infusion studies as well as studies with novel unimolecular dual GIP/GLP-1 receptor agonists.

## 2. GLP-1 (Glucagon-Like Peptide 1)

### 2.1. Physiology of GLP-1

Glucagon and the glucagon-like peptides (GLP-1 and glucagon-like peptide 2 (GLP-2)) originate from tissue specific processing of proglucagon (their common precursor encoded by the glucagon gene) in the pancreas and the gastrointestinal tract, respectively [22]. Secretion of GLP-1 from intestinal L cells is meal-related [23] and occurs in response to the presence of nutrients in the lumen of the intestinal tract [24,25]. Due to the action of the ubiquitous enzyme dipeptidyl peptidase 4 (DPP-4), GLP-1 is rapidly degraded (*T*_½_ 1–2 min) to an inactive metabolite [26]. Active GLP-1 interacts with a specific seven transmembrane G protein-coupled receptor, the GLP-1R, which in humans is known to be expressed in the pancreas, gastrointestinal tract, kidneys, heart, and the central nervous system [27,28,29,30]. Stimulation of the GLP-1R results in activation of adenylate cyclase, which leads to an increase in cellular concentrations of cyclic adenosine monophosphate (cAMP), activation of protein kinase A (PKA) and the cAMP-binding protein, Epac [31,32]. There are two isoforms of Epac, Epac1 and Epac2 [33]. Epac2 is the main isoform present in the pancreas and is involved in GLP-1-mediated potentiation of glucose-induced insulin secretion [34,35]. The role of Epac1 in the pancreas is less clear, but it has been suggested to be involved in the maintenance of beta cell mass and the production of insulin production [36]. As alluded to above, the GLP-1R is widely distributed, and thus, the biological effects of GLP-1 are manifold. Important physiological effects of GLP-1 include enhancement of the insulinotropic response of beta cells to intake of nutrients, reduced gastrointestinal secretion and motility, as well as induction of satiety and ensuing reduction of food intake [37,38,39,40,41,42,43]. Sparse evidence from animal studies suggests that in addition to the well-known effect of GLP-1 on pancreatic insulin secretion, GLP-1R signaling in the brain may also promote insulin secretion and thereby glucose tolerance [44,45]. Importantly, GLP-1 is a potent inhibitor of pancreatic glucagon secretion [46]. The glucagonostatic effect of GLP-1 is dependent on the plasma glucose concentration and GLP-1 inhibits glucagon secretion during euglycemia and hyperglycemia but does not affect glucagon secretion when glucose levels are in the hypoglycemic range (≤3.7 mmol/L) [18]. Blocking the effects of endogenous GLP-1 using the GLP-1R antagonist exendin (9–39)NH_2_ in non-diabetic individuals prior to ingestion of a carbohydrate-rich drink significantly increases postprandial glucagon secretion [47]. Furthermore, physiological doses of exogenous GLP-1 suppress glucagon secretion in non-diabetic individuals during a three-step glucose clamp from fasting plasma glucose to plasma glucose levels of 6 and 7 mmol/L respectively, mimicking postprandial glucose excursions [48]. While there is substantial support for the glucose-dependent glucagonostatic effect of GLP-1, the mechanism(s) by which GLP-1 inhibits glucagon secretion remains uncertain. At least four hypothesizes have been proposed. 1) The intra-islet hypothesis suggests that GLP-1 mediates glucagon suppression via a paracrine mechanism. As the beta cell secretory response to high levels of blood glucose is augmented by GLP-1, so is the paracrine inhibition of glucagon secretion by beta cell secretory products [49]. This hypothesis also embodies the lack of a glucagonostatic effect of GLP-1 observed during hypoglycemia when GLP-1 has no insulinotropic effect. The fact that GLP-1 strongly inhibits glucagon secretion in individuals with type 1 diabetes [50,51,52] nevertheless suggests that other mechanisms are needed to fully account for the modulatory effect of GLP-1 on glucagon secretion. 2) Another hypothesis behind the inhibitory effect of GLP-1 on glucagon is through a direct action of GLP-1 on GLP-1R on the alpha cell. This seems evident in rodents [53], but the presence of the GLP-1R on human alpha cells is a matter of debate. Reports range from complete lack of the GLP-1R in the human alpha cells [54] to the presence of a low number of receptors [55,56], which may be enough to mediate the glucagonostatic effects of GLP-1 [56]. However, specificity issues of commonly used GLP-1R antibodies cast uncertainty upon these results [57,58,59]. In a recent study applying an antibody with improved specificity, human alpha cells were not found to be GLP-1R positive, thus questioning this theory [60]. 3) Neighboring delta cells were reported to express a high density of GLP-1R when the abovementioned antibody was applied [60], supporting the theory that GLP-1-mediated glucagon suppression involves the release of somatostatin from pancreatic delta cells (Figure 1). GLP-1 is a strong mediator of somatostatin release from delta cells [61,62,63,64], and blocking the effects of somatostatin with a somatostatin receptor 2 antagonist diminishes the inhibitory action of GLP-1 in the rat pancreas [61,64]. However, whether this is also the case in humans remains to be established. 4) Finally, sparse evidence suggests that part of the modulatory effect of GLP-1 on insulin and glucagon secretion may be mediated by the parasympathetic nervous system [65,66,67], but the importance of this mechanism in humans requires further substantiation.

Based on a few reports [68,69], the endogenous GLP-1 response to meal ingestion was for years thought to be reduced in patients with type 2 diabetes. This turned out not to be the case [70,71], but spurred investigations of exogenous GLP-1 administration in patients with type 2 diabetes. Interestingly, insulinotropic but not glucagonostatic effects of GLP-1 are significantly reduced in patients with diabetes [72,73]. However, incretin activity can be observed with administration of supraphysiological levels of GLP-1RAs [72]. This, combined with the satiety-promoting and food intake-reducing effects of GLP-1 and its glucose-dependent glucagonostatic effect has paved the way for the development of GLP-1RAs for the treatment of type 2 diabetes [74].

### 2.2. GLP-1R (Glucagon-Like Peptide 1 Receptor) Agonism in Type 2 Diabetes

The pharmacological advantages of GLP-1RAs in type 2 diabetes include correction of fasting and postprandial hyperglycemia (without increased risk of hypoglycemia due to the glucose-dependency of the insulinotropic and glucagonostatic effects of GLP-1; only operating at plasma glucose levels above 4–5 mmol/L) and weight loss. Furthermore, GLP-1RAs reduce systolic blood pressure (most likely due to a natriuretic effect of GLP-1 in the kidneys and body weight loss [75]) and postprandial lipids. Despite the fact that treatment with GLP-1RAs also increases heart rate by 2–8 bpm, cardiovascular outcomes trials have shown that treatment with some GLP-1RAs (liraglutide, albiglutide, semaglutide, and dulaglutide) protect against major adverse cardiovascular events in high-risk patients with type 2 diabetes [76]. Additionally, data from these trials also suggest that GLP-1RAs have renoprotective effects [74,77,78].

The glucagonostatic effect of GLP-1 seems equally important as its insulinotropic action for the glucose-lowering effect of exogenous GLP-1 in patients with type 2 diabetes [79]. Several clinical trials have reported that the GLP-1RAs exenatide, liraglutide, dulaglutide and semaglutide decrease fasting and/or postprandial glucagon secretion [80,81,82,83,84]. However, there are some inconsistencies regarding the long-term effects of the different GLP-1RAs on glucagon secretion. In the LIBRA trial, initial treatment with liraglutide decreased glucagon levels [85], but glucagon responses to oral glucose were increased after 12 weeks of treatment [86]. Furthermore, the AWARD-3 trial reported a significant decrease in fasting glucagon after 26 weeks of dulaglutide treatment when compared to treatment with metformin, but no difference between dulaglutide and metformin was observed after 56 weeks [87]. One may speculate that chronic treatment with GLP-1RAs perhaps decreases alpha cell sensitivity to GLP-1 over time [88]. As a parallel, in vitro studies suggest that rat beta cells are desensitized to GLP-1 during prolonged exposure to GLP-1 [89]. Thus, whereas the glucose-dependent glucagonostatic effect of GLP-1 is important for its acute glucose-lowering effect, uncertainty remains with respect to the contribution of this mechanism to the long-term glucose-lowering effect of GLP-1RAs.

## 3. GIP (Glucose-Dependent Insulinotropic Polypeptide)

### 3.1. Physiology of GIP

GIP is secreted from intestinal K cells in response to the ingestion of nutrients, especially fat and carbohydrates [25]. GIP is similarly to GLP-1 inactivated by DPP-4 resulting in a short half-life of 4–7 min [90,91]. The fasting concentrations are usually below 20 pmol/L but may increase to hundreds of pmol/L in response to a meal dependent on the size of the meal [92]. GIP interacts with the GIPR, which, like the GLP-1R, is a seven transmembrane G protein-coupled receptor. The main cellular pathway involved in GIPR signaling involves an increase in intracellular cAMP concentration and activation of PKA [93,94]. The GIPR is expressed in many tissues including the pancreas, adipose tissue, gastro-intestinal tract, heart, pituitary gland, adrenal cortex, and the brain [95]. This indicates pleiotropic functions, and GIP has been suggested to mediate a reduction in bone resorption [96], increase triacylglyceride deposition in adipose tissue [97] and increase blood flow to the adipose tissue and intestines [97,98]. Interestingly, GIPR knockout protects against hypothalamic leptin resistance in diet-induced obese mice [99]. This suggests obesity-promoting effects of naturally occurring GIP, but contrasting studies have questioned this view [100]. In non-diabetic individuals, GIP potentiates postprandial insulin secretion, but this effect is severely diminished or even absent in people with type 2 diabetes [20,72,101]. Similarly to GLP-1, the insulinotropic effect of GIP is glucose-dependent, i.e., the effect is absent at plasma glucose less than 4–5 mmol/L [19,102]. Interestingly, GIP acts glucagonotropically when administered at euglycemia or hypoglycemia [19] but not during hyperglycemia [19,48]. Due to these inverse glucose-dependent actions on insulin and glucagon secretion, GIP has been suggested to act as a physiological bifunctional stabilizer of blood glucose [19]. For years, the lack of a suitable GIPR antagonist has limited the study of the physiological effects of endogenous GIP. But recently a naturally occurring GIP fragment (GIP(3–30)NH_2_) has been validated as a GIPR antagonist in humans and used to delineate the effects of endogenous GIP in humans [103]. Using GIP(3–30)NH_2_ and the GLP-1R antagonist exendin (9–39)NH_2_, alone and combined, it was shown in healthy individuals that the plasma glucose-lowering effects of endogenous GIP and GLP-1 after meal ingestion are additive and that endogenous GIP has a greater effect on postprandial glucose excursions than endogenous GLP-1. Interestingly, it was also shown that the natural suppression of glucagon secretion occurring after an OGTT was significantly attenuated by exendin(9–39)NH_2_, but remained unaffected by GIP(3–30)NH_2_. Thus, GIP potentiated insulin secretion with no effect on glucagon secretion in a postprandial setting with hyperglycemia (plasma glucose ~5 to ~10 mmol/L). Our group recently infused GIP(3–30)NH_2_ during a liquid meal test in 10 patients with type 2 diabetes [101]. Here, the insulinotropic effect of endogenous GIP was found to be modest, while glucagon levels were unaffected by the antagonist. These observations are in line with the bifunctional role of GIP on insulin and glucagon secretion in non-diabetics and the notion of defective potentiation of insulin secretion by GIP in patients with type 2 diabetes. The role of endogenous GIP in the fasting state remains to be investigated, and hopefully future studies applying GIP(3–30)NH_2_ will expand our knowledge of the role of GIP in normal physiology and in the pathophysiology of type 2 diabetes.

In rodents, the molecular basis for the glucagonotropic action of GIP on alpha cells has been shown to closely resemble the mechanism of GIP on beta cells, i.e., via an increase in intracellular cAMP concentration and activation of PKA (Figure 1) [104]. Studies in the perfused rat pancreas have confirmed that GIP stimulates glucagon secretion directly, while GLP-1 inhibits glucagon secretion in a somatostatin-dependent paracrine mechanism [61]. Human alpha cells have been found to express the GIPR [21]. Upon stimulation with GIP, both cAMP generation and glucagon secretion increased in a concentration-dependent manner which is supportive of a direct action of GIP on alpha cells [21].

### 3.2. GIPR (Glucose-Dependent Insulinotropic Polypeptide Receptor) Agonism in Type 2 Diabetes

Contrary to GLP-1RAs, selective GIPR agonists (GIPRAs) are not available for the treatment of type 2 diabetes. While the secretion of GIP in patients with type 2 diabetes appears to be normal [105], the insulinotropic ability of GIP is severely reduced at both physiological [106] and pharmacological concentrations in these patients [20,72,107]. The diminished insulinotropic effect of GIP may to be due to glucose-induced downregulation of the GIPR [108] and strict glycemic control has been shown to partly restore the beta cell sensitivity to GIP [20,106]. The reduced insulinotropic effect of GIP is considered an important pathophysiological component of type 2 diabetes, which, so far, has rendered the pursuit of GIP monotherapy as anti-diabetic strategy unattractive. Regarding the modulation of glucagon secretion by GIP in type 2 diabetes, some studies have found that in patients with type 2 diabetes, the glucagonotropic action of GIP prevails even during hyperglycemia [21,109]. This may have detrimental effects on postprandial glucose tolerance as exogenous GIP infusion increased postprandial plasma glucose concentrations in type 2 diabetes patients [21]. Observations made by our group during clamping of plasma glucose at different glycemic levels, seem to indicate that the glucagonotropic action of GIP in type 2 diabetes is preserved during periods of hypoglycemia and/or euglycemia [110]. Based on these short-term clinical experiments the prospect of using GIP as an antidiabetic agent seems poor, but the long-term effects of GIPR agonism remains to be investigated.

## 4. GLP-1 and GIP Dual Agonism

### 4.1. In Vitro Studies with GLP-1 and GIP in Combination

In insulinoma cells, co-administration of GIP and GLP-1 has been shown to elicit an additive cAMP response [111]. In line with this, stimulation of rodent beta cells with a combination of GLP-1 and GIP led to greater potentiation of subsequent glucose or GLP-1-induced insulin secretion than stimulation with either peptide alone [89]. In human islets from non-diabetic donors and donors with type 2 diabetes, acute exposure to GIP has been reported to be superior to acute exposure with equimolar amounts of GLP-1 in terms of insulin release [112]. After prolonged exposure, the combination of both incretins had synergistic effects on insulin synthesis, insulin secretion, and expression of genes associated with beta cell differentiation and survival than incubation with only one of the two peptides [112].

### 4.2. Animal Studies with GLP-1 and GIP Co-infusions

Results from animal studies on the potential additive glucose-regulatory effect of GLP-1 and GIP are not consistent. Some studies have observed synergistic effects of dual GLP-1R and GIPR agonism on glucose tolerance, insulin secretion, body weight and food intake in mice [113,114,115]; others have found no additional benefits of dual agonism on these parameters compared to treatment with only one incretin hormone [116]. To our knowledge, glucagon levels have not been reported in these studies. One study examining mice with diet-induced obesity during co-infusions of a GIPRA and a GLP-1RA for 14 days found that co-administration did not potentiate the glucose-lowering effects of either peptide infused alone [117]. However, co-administration augmented reductions in food intake, body weight and fat mass, and this was the rationale for the development of a unimolecular dual incretin receptor agonists (i.e., a single molecular structure with agonistic properties both at the GLP-1R and the GIPR) for the treatment of type 2 diabetes and obesity (this study will be discussed in more details below) [117]. Glucagon levels were not measured. In line with this, it has been shown that co-administration of liraglutide and a novel GIPRA leads to a greater reduction in energy intake and body weight in mice with diet-induced obesity compared to liraglutide alone [118]. Surprisingly, antagonizing the GIPR protected against body weight gain in mice with diet-induced obesity and led to a reduced food intake and resting expiratory exchange rate [119]. These results were replicated in non-human primates with an even more pronounced weight loss [119]. Finally, the combination of GIPR antagonism and a GLP-1RA enhanced the observed weight loss in all animal models [119]. Although GLP-1 and GIP co-administration has almost consistently demonstrated positive effects on body weight, it is paradoxical that both GIPR agonism and antagonism seem to enhance the effects of GLP-1R agonism. This highlights that the mechanisms mediating any additive effect of GIP and GLP-1 in terms of body weight reduction are not well-described; and particularly the role of glucagon in these preclinical findings remain obscure.

### 4.3. Human Studies with GLP-1 and GIP Co-infusions

The effects of short-term GLP-1 and GIP co-infusion have been investigated in both non-diabetic individuals and patients with type 2 diabetes. The separate and combined effects of GLP-1 and GIP were investigated by Nauck et al. in 8 non-diabetic subjects during intravenous isoglycemic glucose infusions (IIGI) mimicking a preceding OGTT [120]. The infusion rates of GIP and GLP-1 (1 and 0.3 pmol/kg/min, respectively) were chosen to reach plasma concentrations identical to the concentrations of the endogenous hormones during the OGTT. The combined effects of the incretins were greater than the separate effects in terms of insulin secretion, but infusion of GLP-1 and/or GIP had no additional effect on glucagon secretion compared to the IIGI [120]. Elahi et al. examined the separate and combined effects of GLP-1 and porcine GIP in 6–7 non-diabetic individuals during a hyperglycemic clamp (5.4 mmol/L above fasting plasma glucose) [121]. The infusion rates were 1.5 and 2 pmol/kg/min for GLP-1 and GIP, respectively. The co-infusion significantly inhibited glucagon secretion and potentiated insulin secretion. These effects were superior compared to infusion of either peptide alone [121]. Daousi et al. investigated the effects of mono-infusions and co-infusion of GLP-1 (1 pmol/kg/min) and GIP (2 pmol/kg/min) during a 4-h intravenous glucose infusion in 6 non-diabetic men and 6 men with type 2 diabetes [122]. In the non-diabetic men, the co-infusion significantly increased insulin response to glucose compared to mono-infusion of either hormone alone. This was not the case in the patients with type 2 diabetes, which is in line with the generally reduced insulinotropic effect of GIP in individuals with type 2 diabetes as previously described. Unfortunately, glucagon responses were not reported in this study. Nevertheless, scrutinizing the plasma glucose excursions, it may seem that GIP alone had a moderate hyperglycemic effect in both groups. Furthermore, co-administration of GLP-1 and GIP resulted in glucose excursions identical to those observed with mono-infusion of GLP-1 despite a numerically greater insulin response during co-infusion. One may speculate that GIP-mediated glucagon secretion underlies this phenomenon. Our group has studied the separate and combined effects of GIP, GLP-1 and GLP-2 (the latter known to be glucagonotropic but without any notable effect on insulin secretion [123]) on glucagon secretion in 10 patients with type 2 diabetes [109]. After an initial OGTT, the participants completed four IIGIs with infusions of either GIP (4 pmol/kg/min), GLP-1 (0.6 pmol/kg/min), GLP-2 (1 pmol/kg/min) or the combination of all three peptides. The infusion rates were chosen to mimic physiological plasma concentrations of the peptides in the portal circulation. While GLP-1 suppressed glucagon secretion and GIP promoted the release of glucagon, we observed no changes in glucagon secretion during co-infusion, suggesting that the bidirectional effects of GLP-1 and GIP/GLP-2 on glucagon secretion nullify each other. Furthermore, insulin secretion was equally potentiated by infusion of GLP-1 alone or in combination with GIP (and GLP-2), but not GIP alone, confirming the reduced insulinotropic effect of GIP in these patients. These findings were confirmed when Mentis et al. investigated whether GIP could potentiate the insulinotropic or glucagonostatic effects of GLP-1 in type 2 diabetes without concomitant glucose infusion [107]. After an overnight fast, 12 patients with type 2 diabetes received infusions of GIP (4 pmol/kg/min), GLP-1 (1.2 pmol/kg/min) or the combination of both for 6 h. In this setting, GIP did not enhance the insulin response to GLP-1 during co-infusion, which also resulted in blood glucose levels equal to infusion of GLP-1 alone. Furthermore, GLP-1-induced inhibition of glucagon secretion was reduced during co-infusion [107]. Recently, our group investigated the effects of combined GIP and GLP-1 infusion in seventeen overweight/obese individuals during 4 different IIGIs (mimicking a preceding OGTT) with co-infusion of either saline, GIP (4 pmol/kg/min), GLP-1 (1 pmol/kg/min) or GIP + GLP-1 (4 and 1 pmol/kg/min, respectively) [124]. Compared to IIGI + saline, IIGI + GLP-1 suppressed glucagon secretion, whereas the glucagonotropic response during IIGI + GIP + GLP-1 did not differ from IIGI + saline, suggesting opposing actions of GLP-1 and GIP on glucagon secretion in these non-diabetic overweight/obese men. The insulin response was similar during infusions of GLP-1 and GLP-1 + GIP and far greater than during infusion of GIP alone. This may imply that overweight/obese men, similarly to patients with type 2 diabetes, have reduced beta cell sensitivity to GIP. In fact, this study was designed to evaluate the effect of GIP—alone and in combination with GLP-1—on ad libitum meal intake; only the infusion of GLP-1 alone resulted in a significant decrease in energy intake compared to saline. Unpublished data from our group (Bergmann et al., unpublished) suggest that individuals with type 2 diabetes who are treated with a long-acting GLP-1RA have no additional benefit of high-dose exogenous GIP. Twenty-two patients (all treated with metformin and a long-acting GLP-1RA) received 5-h infusions of GIP (6 pmol/kg/min) or saline. Initially, a standardized liquid mixed meal test was performed, and an ad libitum meal was served at the end of the infusion for evaluation of energy intake. GIP had no influence on energy intake, appetite, energy expenditure, gastric emptying, gallbladder motility or plasma lipids compared to placebo. However, glucagon secretion was augmented during GIP infusion and plasma glucose excursions were significantly greater compared to a placebo.

### 4.4. Unimolecular Dual GLP-1 and GIP Receptor Agonists

After the demonstration of the synergistic effects of GLP-1 and GIP co-administration in mice, a series of peptides with balanced agonism at both the GLP-1R and GIPR was derived from a glucagon-core [117]. One such unimolecular dual-agonist, RG7697, was found to display balanced and potent activity at GIPR and GLP-1R and minimal activity at the glucagon receptor in vitro. RG7697 is believed to bind to only one receptor at a time, resulting in less GLP-1R occupancy at equimolar doses to those of selective GLP-1RAs and ultimately fewer gastrointestinal side effects, which is considered the dose-limiting step of the GLP-1RAs. It is still not clear to which degree GLP-1R and GIPR activation, respectively, contribute to the activity of unimolecular dual agonists and in vivo potency of dual agonists at the incretin receptors remains to be determined. At any rate, when administered at equimolar doses to diet-induced obese mice, RG7697 led to a greater weight loss while demonstrating superior insulinotropic and antihyperglycemic effects after a glucose challenge compared to selective GLP-1RAs in *db*/*db* mice, lean rats, Zucker diabetic fatty rats, and cynomolgus monkeys [117]. No glucagon data were published from this study.

In 18 non-diabetic humans, RG7697 was infused during a glucose infusion with positive effects on glucose control paralleled by an increased glucose-induced insulin response [117]. However, there was no direct comparison with a GLP-1RA. Furthermore, in 44 patients with type 2 diabetes, RG7697 dose-dependently reduced HbA1c during six weeks of treatment but, again, no comparison was made to a selective GLP-1RA [117]. To our knowledge, no glucagon measurements have been reported from either of these human studies.

In another study, the effects of ascending doses of RG7697 administered subcutaneously to 38 non-diabetic subjects over 96 h were evaluated [125]. Assessed by a meal tolerance test after treatment, RG7697 dose-dependently reduced maximal plasma glucose and insulin concentrations by up to ~50% compared to a meal tolerance test at baseline [125]. This was not accompanied by any changes in postprandial glucagon levels even at the highest dosing. With no comparison to a selective GLP-1R, it is not clear if the benefits of dual-agonism observed here surpassed those of pharmacological GLP-1R activation alone.

In two cohorts of ~40 patients with type 2 diabetes, RG7697 reduced fasting, postprandial and 24-h glucose levels as well as HbA1c dose-dependently [126,127]. Postprandial insulin levels decreased after treatment with RG7697 while postprandial glucagon levels were reported to be slightly reduced [126] or not reported at all [127]. Treatment with RG7697 was also associated with weight loss, but this was comparable to the weight loss observed with placebo treatment [126,127]. RG7697 was not directly compared with a selective GLP-1RA so whether RG7697′s GIPR-stimulating effect contributes to the abovementioned effects remains unclear.

A novel dual GIPR/GLP-1R agonist, tirzepatide (also known as LY3298176), has been developed for once-weekly administration [128]. In vitro, the affinity of tirzepatide is comparable to natural GIP for the GIPR and ~5-fold weaker than natural GLP-1 for the GLP-1R. Additionally, tirzepatide potently stimulates cAMP accumulation in recombinant cell lines expressing the GIPR or the GLP-1R, but has minimal activity on the glucagon receptor [128]. In mice, glucose tolerance after intraperitoneal glucose infusion was improved by tirzepatide [128]. Compared to a selective GLP-1RA, tirzepatide provided better weight loss and a superior reduction in food intake with increased energy expenditure in mice with diet-induced obesity [128]. Glucose excursions during OGTT were decreased dose-dependently with tirzepatide and also significantly decreased with dulaglutide, and insulin levels during OGTT were similar after treatment with either compound [128]. Glucagon levels during OGTT were not reported.

Ascending doses of tirzepatide (0.5–15 mg once-weekly) over 4 weeks was compared to placebo in 25 normal glucose tolerant subjects and 42 patients with type 2 diabetes [128]. The GLP-1RA dulaglutide (1.5 mg once-weekly) was administered to a small group of 4 non-diabetic individuals as a control. Tirzepatide dose-dependently reduced fasting glucose but not fasting insulin in non-diabetic individuals. Oral glucose tolerance was significantly improved across all doses of tirzepatide and dulaglutide but with no effect on insulin responses to OGTTs. Subjects receiving the highest dosing of tirzepatide reduced body weight by ~4 kg after 4 weeks of treatment [128]. In patients with type 2 diabetes, tirzepatide was shown to reduce fasting glucose and insulin levels dose-dependently [128]. Oral glucose tolerance was significantly improved and accompanied by enhanced OGTT-induced insulin secretion. A dose and time-dependent weight loss was also observed, although with a smaller effect size when compared to the non-diabetic subjects. To our knowledge, no glucagon outcomes have been reported from this trial. The efficacy of tirzepatide was later evaluated in a larger group of patients with type 2 diabetes (*n* = 316) [129]. Participants were randomly assigned to receive either tirzepatide (1, 5, 10 or 15 mg), dulaglutide (1.5 mg — currently the highest recommended dose) or placebo for 26 weeks (~50 participants in each group). To help improve tolerability, patients assigned to the 10 mg group received 5 mg for 2 weeks and 10 mg for the rest of the study. Patients assigned to the 15 mg group received 5 mg for 2 weeks, then 10 mg for two weeks and 15 mg for the remaining duration of the study. At 26 weeks, tirzepatide decreased HbA1c dose-dependently and without plateau (−1.06, −1.75, −1.89 and −1.94% for 1, 5, 10 and 15 mg, respectively) [129]. The reduction in HbA1c by dulaglutide was −1.21%. Body weight was reduced with tirzepatide (−0.9, −4.8, −8.7 and −11.3 kg for 1, 5, 10 and 15 mg, respectively) compared to a −2.7 kg reduction in weight by dulaglutide [129]. Tirzepatide, like dulaglutide, reduced fasting and postprandial plasma glucose. Interestingly, fasting glucagon decreased dose-dependently with tirzepatide, while fasting insulin decreased at all doses above 1 mg [129]. Dulaglutide (1.5 mg) reduced fasting glucagon and increased fasting insulin in a manner comparable to 1 mg tirzepatide. At the highest dosing, tirzepatide surpassed dulaglutide in number of adverse events, especially gastrointestinal adverse events (66.0% of individuals treated with tirzepatide 15 mg and 42.6% of individuals treated with dulaglutide 1.5 mg), but also in total number of hypoglycemic episodes (7.5% of participants receiving 15 mg tirzepatide versus 3.7% of participants receiving dulaglutide) [129]. To circumvent this issue, an adverse event-lowering titration strategy for tirzepatide has been reported [130]. However, the alternative three-step dose escalation regimen for 15 mg tirzepatide did not seem to effectively lower the overall incidence of gastrointestinal side effects compared to the previously reported study [129]. The incidence of hypoglycemic episodes also increased to ~16% of participants receiving 15 mg tirzepatide. Although tirzepatide treatment was still associated with a promising weight loss (~5.6 kg after 12 weeks), the high rate of gastrointestinal side effects did cast some uncertainty upon the clinical potential of tirzepatide. From a mechanistic point of view, it is interesting to note that tirzepatide has been shown to delay gastric emptying in a manner similar to the selective GLP-1RA semaglutide with tachyphylaxis after prolonged treatment [131]. Furthermore, tirzepatide was also found to regulate human adipocytes via GIPR specific activity, perhaps providing some of the mechanistic basis for its pharmacological body weight-lowering effect [132]. Taken together, the abovementioned trials suggest that dual incretin receptor agonists may constitute effective HbA1c-lowering and body weight-reducing agents in patients with type 2 diabetes. But these effects need to be considered in light of the frequent gastrointestinal adverse events at the most efficacious doses.

Whether some of the effects of dual incretin receptor agonism are mediated via glucagon remains mostly unknown. Many of the aforementioned studies did not report any glucagon outcomes, but some trials found a decrease in postprandial [126] or fasting [129] plasma glucagon concentrations, while others reported no changes in postprandial glucagon secretion following dual incretin receptor agonist treatment [125]. One study found that in patients with type 2 diabetes, tirzepatide decreased glucagon concentrations despite concomitant lowering of plasma glucose, much like a GLP-1RA, but with greater magnitude [129]. This is surprising, given that the GIP component of tirzepatide is glucagonotropic in patients with type 2 diabetes as would be expected from co-infusion studies with GIP and GLP-1. This raises questions regarding the effect of GIPR activation in addition to chronic GLP-1R agonism: Is the GIP component active in humans or is it the effect of potent GLP-1R activation that is observed? And if it is active, what is the contribution of the GIP component to the mode of action of these drugs? Interestingly, when comparing the effects of 1.5 [128] or 1 mg tirzepatide [129] on body weight and glycaemia with the effects of 1.5 mg dulaglutide, they were more or less identical, but with fewer gastrointestinal adverse effects. Furthermore, the relatively high incidence of hypoglycemic episodes may suggest that the GIP component is not active and therefore does not protect against hypoglycemia. It is difficult to delineate the individual contribution of the GLP-1 and GIP components to the effects of the dual agonists. Comparing these drugs to long-acting selective GLP-1 and GIP agonists separately would help define the contribution of each incretin component to the effects of unimolecular dual-agonism. In any case, the dual incretin receptor agonists appear to be better tolerated than their selective GLP-1R agonistic counterparts at equimolar doses, which may allow for higher dosing than with GLP-1RAs. Whether the apparently enhanced tolerability is due to the GIP component or other factors remains to be determined. Glucagon, with its pleiotropic functions on hepatic glucose production, energy homeostasis and body weight, is arguably an important endocrine factor to consider when delineating the effects of dual GLP-1R/GIPR agonism. Given that GLP-1 is a potent inhibitor of glucagon in patients with type 2 diabetes [109], it not surprising that glucagon concentrations seem to diminish following dual incretin receptor agonist treatment—although this observation is far from consistent and supported by only two studies [126,129]. However, considering that GIP acts glucagonotropically in patients with type 2 diabetes [21,109] the GIP component of dual incretin receptor agonists might counteract GLP-1-mediated inhibition of glucagon secretion, leading to a neutral effect on glucagon levels. Indeed, one study found that postprandial glucagon levels were unaffected by short-term treatment with a dual incretin receptor agonist [125]. These are speculative interpretations and future studies are needed to delineate the role of glucagon in the mode of action of this emerging treatment strategy for obesity and type 2 diabetes.

## 5. Conclusions and Perspectives

Recent studies have demonstrated beneficial effects on glycemic control and body weight with dual GIPR/GLP-1R agonists. Hyperglucagonemia is a common phenomenon in type 2 diabetes and—via glucagon’s stimulatory effect on endogenous glucose production—it contributes to the core pathophysiological trait of these patients namely hyperglycemia. Also, glucagon seems to regulate energy homeostasis and body weight. As both incretin hormones are known to modulate glucagon secretion (in opposite ways and at different glucose levels), pharmacological dual incretin receptor agonism may have effects on glucagon secretion that are of clinical importance.

It is difficult to disentangle the individual effects of GLP-1R and GIPR activation by the emerging unimolecular dual incretin receptor agonist compounds. The possibility exists that dual incretin receptor agonists simply work by potent activation of the GLP-1R; with any GIPR activity being clinically unimportant. At equimolar doses, the unimolecular dual incretin receptor agonist tirzepatide has effects that are strikingly comparable to dulaglutide. The individual contribution of the GIPR and GLP-1R to the pharmacological effects of long-term dual incretin receptor agonism could be determined by individual and combined administration of selective mono GLP-1R and mono GIPR antagonists or by comparison with long-term selective GLP-1 and GIP agonists.

Preclinical studies illustrate the paradox that both GIPR agonism and antagonism compliments GLP-1R activity. It may be speculated that long-term GIPR agonism may result in GIPR downregulation to such a degree that it effectively becomes universal GIPR antagonism. This could contribute to body weight loss, since GIP seems to increase blood flow to adipose tissue and facilitate triacylglycerol deposition in adipocytes. If this applies to dual incretin receptor agonists, antagonizing the GIPR may compliment the GLP-1R-mediated inhibition of glucagon secretion, which would ultimately lower endogenous glucose production and plasma glucose.

Interestingly, suppression of glucagon activity by the GLP-1 component of dual incretin receptor agonists may not be clinically beneficial after all, as studies investigating GLP-1, GIP and glucagon receptor tri-agonists have shown potential in reducing body weight and hyperglycemia [133,134,135,136]. Future studies will need to delineate the role of both glucagon, GLP-1, and GIP in the mode of action of glucagon and incretin-based drug candidates.

## Figures and Tables

**Figure 1 ijms-20-04092-f001:**
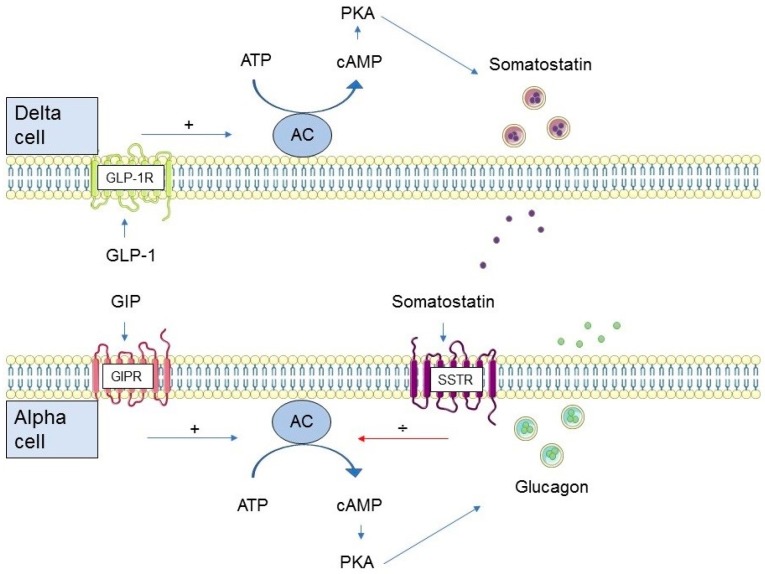
Proposed mechanisms of glucose-dependent insulinotropic polypeptide (GIP) and glucagon-like peptide 1 (GLP-1) modulation of alpha cell glucagon secretion in humans. AC: adenylate cyclase, ATP: adenosine triphosphate, cAMP: cyclic adenosine monophosphate, PKA: protein kinase A, GIPR: GIP receptor, GLP-1R: GLP-1 receptor, SSTR: somatostatin receptor.

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
