# Peer review of "The Effects of Dual GLP-1/GIP Receptor Agonism on Glucagon Secretion—A Review"

_ijms, 2019, doi:10.3390/ijms20174092_

Round 1

Reviewer 1 Report

The authors gave a very comprehensive review on the effect of GLP-1/GIP receptor agonism on  glycemic control, body weight, glucagon secretion, and insulin levels. Below are my comments.

1.     The title focuses on the effect of dual agonism on glucagon secretion, while glucagon data are not record/reported in many of the animal and human studies reviewed. I suggest the authors to change glucagon secretion to glycemic control.

2.     Line 231. Are there any co-infusion studies done with diabetic animal models?

3.     Line 300. Please briefly explain what does “unimolecular” mean.

4.     Line 306. “At any rate, when administered at equimolar doses to rats”. Is it administered to “rats” or “diet-induced obesity mice, db/db mice, Zucker diabetic fatty rats and cynomolgus monkeys”

5.     Line 338. Is it reported in this study how tirzepatide compared to GLP-1RA in terms of glycemic control/glucose tolerance/insulin level/glucagon level?

6.     Line 352. How is dulaglutide dose determined in this study?

Author Response

The title focuses on the effect of dual agonism on glucagon secretion, while glucagon data are not record/reported in many of the animal and human studies reviewed. I suggest the authors to change glucagon secretion to glycemic control.

AUTHOR REPLY: The special issue of IJMS focuses on glucagon biology and pharmacology. In that context, this review is relevant due to its focus on the mechanistic basis of modulation of glucagon secretion by the incretin hormones and the discussion of the available glucagon data from trials with dual agonism. We suggest keeping the title to preserve the glucagon-centric nature of the review.

Line 231. Are there any co-infusion studies done with diabetic animal models?

AUTHOR REPLY: Five of the reviewed animal trials were conducted with diabetic mice or rats (Irwin et al. Diabetes Metab. Res. Rev. 2007; Gault et al. Clin. Sci. Lond. Engl. 1979 2011; Irwin et al. J. Pept. Sci. Off. Publ. Eur. Pept. Soc 2007; Finan et al. Sci. Transl. Med. 2013; Nørregaard et al. Diabetes Obes. Metab. 2018). To our knowledge, these are the available co-infusion studies in diabetic animal models.

Line 300. Please briefly explain what does “unimolecular” mean.

AUTHOR REPLY: It means a single molecular structure with agonistic properties both at the GLP-1R and the GIPR. This has been clarified in the revised version of the manuscript (lines 241-242).

Line 306. “At any rate, when administered at equimolar doses to rats”. Is it administered to “rats” or “diet-induced obesity mice, db/db mice, Zucker diabetic fatty rats and cynomolgus monkeys”

AUTHOR REPLY: Thanks for pointing this out. The effects of liraglutide dual incretin receptor agonism on weight loss was only observed in DIO mice (and not rats, as wrongly stated in the current manuscript). The effect on insulin secretion and glycemia after a glucose challenge was observed in db/db mice, lean rats, ZDF rats and cynomolgus monkeys (but not in DIO mice). This has been clarified in the revised manuscript (lines 326-328).

Line 338. Is it reported in this study how tirzepatide compared to GLP-1RA in terms of glycemic control/glucose tolerance/insulin level/glucagon level?

AUTHOR REPLY: It was reported that glucose AUC after OGTT was significantly reduced with both tirzepatide and dulaglutide with no difference in insulin AUC. There were no glucagon data reported. This has been added to the revised version of the manuscript (please see lines 357-360).

Line 352. How is dulaglutide dose determined in this study?

AUTHOR REPLY: 1.5 mg once weekly is the highest dosing regimen for dulaglutide according to current guidelines. This dose was chosen to compare tirzepatide with a high standard GLP-1RA treatment strategy. We have clarified this in the revised version (please see line 374).

Reviewer 2 Report

This review titled “The Effects of Dual GLP-1/GIP Receptor Agonism on Glucagon Secretion” and authored by Mathiesen et al. summarizes and discusses coordinated effects on glucagon secretion by dual agonism via acting on receptors of two incretin hormones, glucagon-like peptide 1 (GLP-1) and glucose-dependent insulinotropic polypeptide (GIP).  The authors reviewed its glycemic controlling effects by a GLP-1/GIP receptor agonist on preclinical studies and clinical studies using diabetic and obese patients. The authors first reviewed the mechanisms of GLP-1 and GIP underlying pancreatic glucagon secretion; then the effects of coadministration of GLP-1 and GIP on glucagon secretion; and finally discuss therapeutic roles of unimolecular dual GLP-1/GIP receptor agonists. This is an important topic with high interest in the field.

(1) My major concern is the organization of this review, which could be fixed in a revision.

The reviewer started with two paragraphs without a section title. I suggest to add a section title “introduction” to these first two paraphs, which serves general introduction (section 1) for this review.

The second section will discuss GLP-1: 2.1 begin with “Physiology of GLP-1”, then 2.2 “GLP-1R agonism in type 2 diabetes” (current 1.1).

The third section is to similarly discuss GIP: 3.1 begin with “Physiology of GIP” (current 1.2), then 3.2 “GIPR agonism in type 2 diabetes” (current 1.3).

The fourth section is the merit part of this review that discusses dual effects of GLP-1 and GIP in glucose-induced insulin secretion. A background introduction of their interaction may be added before discussing literatures (4.1), followed by in vitro (4.2), in vivo animal studies (4.3) and human studies (4.4).

The last section (section 5) is the conclusions and perspectives (current section 2).

(2) The other major comment is that sex differences in GLP-1 and GIP have been reported in animal and human studies in literature. Adding summary and discussion of available knowledge on sex differences in dual incretin effects of GLP-1 and GIP into this review would improve this review to a next level.

I also have below minor comments and suggestions for the authors when they prepare a revision.

(1) Line 71: the authors discussed GLP-1 binds to its receptor and activate PKA and Epac. The authors could go a little deeper to discuss two subtypes of Epac, Epac1 and Epac2 at pancreatic β cells, and which subtype is activated by GLP-1 and/or GLP-2.

(2) The authors mentioned that GLP-1R and GIPR are found in the central nervous system (lines 68-69 and line 151), the presumable GLP-1 and GIP central effects on glucose regulation need to be added besides their peripheral effects on pancreatic cells.

(3) The dosages and time course of GLP-1 and GIP are specifically mentioned in human studies (for example: lines 240-241). Are the dosages and time courses of GLP-1 and GIP used in vitro and in vivo animal studies relevant to / comparable with those used in human studies?  

This comment also applies to agonists for GLP-1R and GIP1R that their dosages and time course are specified in human studies (for example lines 339-340) but not in in vitro or in vivo animal studies.

Author Response

(1) My major concern is the organization of this review, which could be fixed in a revision.

The reviewer started with two paragraphs without a section title. I suggest to add a section title “introduction” to these first two paraphs, which serves general introduction (section 1) for this review.

The second section will discuss GLP-1: 2.1 begin with “Physiology of GLP-1”, then 2.2 “GLP-1R agonism in type 2 diabetes” (current 1.1).

The third section is to similarly discuss GIP: 3.1 begin with “Physiology of GIP” (current 1.2), then 3.2 “GIPR agonism in type 2 diabetes” (current 1.3).

The fourth section is the merit part of this review that discusses dual effects of GLP-1 and GIP in glucose-induced insulin secretion. A background introduction of their interaction may be added before discussing literatures (4.1), followed by in vitro (4.2), in vivo animal studies (4.3) and human studies (4.4).

The last section (section 5) is the conclusions and perspectives (current section 2).

AUTHOR REPLY: Thanks for these constructive suggestions. The revised version is structured as suggested by the reviewer.

(2) The other major comment is that sex differences in GLP-1 and GIP have been reported in animal and human studies in literature. Adding summary and discussion of available knowledge on sex differences in dual incretin effects of GLP-1 and GIP into this review would improve this review to a next level.

AUTHOR REPLY: Although interesting, to our knowledge no firm conclusions on sex differences regarding the clinical effects of dual incretin receptor agonists can yet be drawn based on current available data.  

I also have below minor comments and suggestions for the authors when they prepare a revision.

Line 71: the authors discussed GLP-1 binds to its receptor and activate PKA and Epac. The authors could go a little deeper to discuss two subtypes of Epac, Epac1 and Epac2 at pancreatic β cells, and which subtype is activated by GLP-1 and/or GLP-2.

AUTHOR REPLY: A characterisation of the two Epac subtypes has been included in the revised version of the manuscript (please see lines 78-82).

The authors mentioned that GLP-1R and GIPR are found in the central nervous system (lines 68-69 and line 151), the presumable GLP-1 and GIP central effects on glucose regulation need to be added besides their peripheral effects on pancreatic cells.

AUTHOR REPLY: A brief summary of the central effects of GLP-1R and GIPR agonism has been added to the revised version of the manuscript (please see lines 86-88 and 168-170).

(3) The dosages and time course of GLP-1 and GIP are specifically mentioned in human studies (for example: lines 240-241). Are the dosages and time courses of GLP-1 and GIP used in vitro and in vivo animal studies relevant to / comparable with those used in human studies?  

This comment also applies to agonists for GLP-1R and GIP1R that their dosages and time course are specified in human studies (for example lines 339-340) but not in in vitro or in vivo animal studies.

AUTHOR REPLY: The dosages and time courses are primarily included for the reader to be able to compare the individual human studies and to evaluate whether the observed effects are physiological or pharmacological. The animal studies (both in vitro and in vivo) are reviewed briefly to provide the rationale for the human studies, which are described extensively and in more detail. Furthermore, the dosages and time courses are not comparable between animals and humans. Therefore, they are included for humans, but not for animals.

Reviewer 3 Report

This is a comprehensive and balanced review of literature, it is well written and presented. I really enjoyed reading this manuscript without any major concerns.

Author Response

Thank you for your kind review of our manuscript. 

Reviewer 4 Report

Thank you for providing me with the opportunity to review your manuscript. The manuscript details The Effects of Dual GLP-1/GIP Receptor Agonism on 3 Glucagon Secretion This topic has clinical significance.

In its present form, the manuscript lacked detail and coherence. I provide the following comments which would strengthen the reporting of your work:

Abstract should be modified because it not clear describe the main objective.

How did you carry out the review? What kind of databases did you use? Why did you use?

What terms did you use and how?

The authors should describe what implications have this review to clinicians?

Author Response

In its present form, the manuscript lacked detail and coherence. I provide the following comments which would strengthen the reporting of your work:

Abstract should be modified because it not clear describe the main objective.

AUTHOR REPLY: We feel that the abstract clearly states the main objective of the review in line 25-28.

How did you carry out the review? What kind of databases did you use? Why did you use?

AUTHOR REPLY: Relevant literature for this review was retrieved from searches in the electronic PubMed database using the following search terms: “incretin”, “GLP-1”, “GIP”, “glucagon” and “dual agonist”. Furthermore, manual reference searches in relevant papers and abstracts from scientific meetings were performed. This has been added to the revised version (lines 59-62).

What terms did you use and how?

AUTHOR REPLY: Please see above.

The authors should describe what implications have this review to clinicians?

AUTHOR REPLY:  This manuscript will offer researchers and clinicians an understanding of incretin physiology and pharmacology and a provide mechanistic insight into future antidiabetic and obesity treatments. This has been added to the revised version of the manuscript (Please see lines 28-30)